# Trust-Gated State Space Models for Budgeted Decisions with Online Risk Control

## Abstract

Deployed predictors increasingly operate with unreliable inputs and strict resource budgets, where labels can be noisy, sensors fail, and conditions drift. We introduce *Trust-Gated State Space Models* (TG–SSM), a compact approach that treats reliability as a first-class control signal. TG–SSM augments a lightweight state-space backbone with gates that modulate input injection, state mixing, and output temperature using on-the-fly reliability features; a small conformal layer then converts probabilities into calibrated *prediction sets* for budgeted decision-making with target risk $1 - \alpha$. On CIFAR-10N (noisy labels), TG–SSM with weighted conformal prediction (WCP) reduces ECE from $0.124$ to $0.048$ while increasing coverage from $0.894$ to $0.905$ at $\alpha = 0.1$ (mean set size $\approx 9.08$). Averaged over CIFAR-10C severities 1–5, TG–SSM+WCP achieves near-nominal coverage ($0.905$) with markedly improved calibration (ECE $0.104$–$0.109$) and compact sets ($\approx 6.74$). On Camelyon17/WILDS (domain shift), a validation-quantile variant attains AUROC $0.949$ with average coverage $0.844$ and set size $\approx 0.989$, while a shift-aware (importance-weighted) variant yields smaller sets ($\approx 0.737$) with AUROC $0.936$. Overall, TG–SSM provides a simple, hardware-efficient recipe for turning uncertain predictions into actionable, budgeted decisions.

## 1 Introduction

Modern learning systems must act under *unreliable data* and *explicit budgets*. Labels may be noisy, sensors intermittently drop values, and deployment conditions drift. In such settings, models are judged not only by accuracy but by whether they can *take actions under uncertainty* while respecting compute or review budgets. We address this by coupling an efficient state-space model with *online risk control* so predictions arrive as calibrated *sets* whose coverage remains near a target level as conditions change.

State space models (SSMs) offer linear-time inference, small memory footprints, and strong predictive quality (Gu et al., 2022; 2020; Tay et al., 2020). However, standard training leaves them *stateless with respect to reliability*: the update rule neither reacts to signs of trouble (e.g., high entropy, abrupt confidence swings) nor communicates calibrated uncertainty. Prior robustness methods harden the model offline or adapt online at compute cost and often do not translate uncertainty into actionable, budget-aware outputs (Zhang & Sabuncu, 2018; Han et al., 2018; Jiang et al., 2018; Chen et al., 2019; Arjovsky et al., 2019; Sagawa et al., 2020; Hendrycks et al., 2020; 2021).

**Thesis.** Reliability should be a *control signal* inside the model. We introduce *Trust-Gated State Space Models* (TG–SSM), which augment a lightweight SSM with small *trust gates* that modulate (i) input injection, (ii) state mixing, and (iii) optional output temperature based on streaming reliability features computed from logits. To make outputs actionable, we add a conformal module that converts probabilities into *prediction sets* and tracks a user-specified risk budget $\alpha$ online (Vovk et al., 2005; Shafer & Vovk, 2008; Vovk et al., 2009; Barber et al., 2021).

**Problem framing.** We cast deployment as a *budgeted decision* problem with risk constraints: minimize decision cost subject to a target miscoverage and limits on compute or human review. The SSM produces reliability-aware scores; the conformal module adjusts set sizes to meet the risk budget; the operator trades coverage for effort along a cost–budget frontier (Elkan, 2001; Bartlett & Wegkamp, 2008; Grubb & Bagnell, 2012).

**Empirical preview.** Experiments (Sec. 5) show: (i) on CIFAR-10N, TG–SSM+WCP reduces ECE by 0.076 absolute while raising coverage to 0.905; (ii) averaged over CIFAR-10C, TG–SSM+WCP improves ECE to 0.104–0.109 at near-nominal coverage (0.905) with compact sets ($\approx 6.74$); (iii) on Camelyon17/WILDS, a validation-quantile variant attains AUROC 0.949 with average coverage 0.844 and set size $\approx 0.989$, while a shift-aware, density-ratio weighted variant further reduces set size to $\approx 0.737$ with AUROC 0.936.

**Contributions.**

1. **Reliability-gated SSM.** A compact SSM whose state update is modulated by *trust gates* derived from on-the-fly reliability statistics.

2. **Online risk control via prediction sets.** A lightweight conformal module that produces *set-valued* predictions and maintains a target miscoverage under gradual shift, supporting budgeted decision-making on-device (Barber et al., 2021; Romano et al., 2019; Barber et al., 2022; Romano et al., 2020).

3. **Deployment-centric evaluation.** A broad evaluation under noisy labels, corruptions, and medical shift, reporting coverage, set size, calibration, and cost–budget frontiers.

4. **Practicality.** The method is simple, hardware-efficient, and complementary to robust training and test-time adaptation (Wang et al., 2021); all components run in a single forward pass with negligible overhead.

## 2 RELATED WORK

**Linear RNNs and State Space Models.** Linear recurrent architectures—including SSMs and linear/causal attention—are competitive, hardware-efficient alternatives to attention-heavy Transformers (Gu et al., 2022; 2020; Tay et al., 2020). Recent models add input- or state-dependent gating (Gu & Dao, 2023; Peng et al., 2023; Wang et al., 2022). Our gates are *reliability-driven*: they modulate computation when evidence appears unreliable.

**Robustness under noise and shift.** Work on label noise includes specialized losses and sample selection (Zhang & Sabuncu, 2018; Han et al., 2018; Jiang et al., 2018; Chen et al., 2019); distribution-shift robustness uses augmentation, invariance, and group-robust objectives (Arjovsky et al., 2019; Sagawa et al., 2020; Hendrycks et al., 2020; 2021). These methods typically output *points*; we complement them with an uncertainty-to-decision layer producing *prediction sets*.

**Calibration and selective prediction.** Post-hoc calibration reduces miscalibration (Guo et al., 2017; Platt, 1999; Zadrozny & Elkan, 2002). Selective prediction and abstention trade accuracy for coverage (Geifman & El-Yaniv, 2019; Chow, 1957). TG–SSM stabilizes logits; the conformal layer enforces a target miscoverage and exposes a budget interface.

**Conformal prediction.** Conformal prediction yields finite-sample valid *set-valued* predictions (Vovk et al., 2005; Shafer & Vovk, 2008); Mondrian (class-conditional) CP improves per-class coverage (Vovk et al., 2009). Online/streaming variants maintain validity via sliding windows and decays (Barber et al., 2021). Under covariate shift, importance weighting adjusts quantiles (Romano et al., 2019; Barber et al., 2022; Romano et al., 2020).

**Budgeted decisions and anytime prediction.** Cost-sensitive learning, reject options, and anytime prediction provide decision-theoretic foundations for budgeted inference (Elkan, 2001; Bartlett & Wegkamp, 2008; Grubb & Bagnell, 2012).

## 3 BACKGROUND

We summarize (i) compact SSM layers, (ii) set-valued prediction via conformal thresholds, and (iii) a budgeted decision view.

## 3.1 STATE SPACE MODELS

Given inputs $\boldsymbol{x}_{1:T}$ and hidden $\boldsymbol{h}_t \in \mathbb{R}^{d_s}$, a linear SSM layer updates

$$\tilde{\boldsymbol{h}}_t = \boldsymbol{A}\,\boldsymbol{h}_{t-1} + \boldsymbol{B}\,\boldsymbol{x}_t, \qquad \boldsymbol{h}_t = (1-\alpha)\,\boldsymbol{h}_{t-1} + \alpha\,\tilde{\boldsymbol{h}}_t, \qquad \boldsymbol{z}_t = \boldsymbol{W}\,\mathrm{LN}(\boldsymbol{h}_t), \qquad (1)$$

with learned $\boldsymbol{A}, \boldsymbol{B}, \boldsymbol{W}$, step size $\alpha \in (0, 1)$, and layer norm. A stack of $L$ blocks with a classifier produces logits $\boldsymbol{z} = \frac{1}{T}\sum_t \boldsymbol{z}_t$ and probabilities $\boldsymbol{p} = \mathrm{softmax}(\boldsymbol{z})$. Modern SSMs parameterize $\boldsymbol{A}$ for linear-time, constant-memory inference (Gu et al., 2022).

**Reliability features.** We compute simple statistics of $\boldsymbol{p}$ to summarize on-the-fly evidence quality:

$$r_t = \big[\, H(\boldsymbol{p}_t),\ \mathrm{margin}(\boldsymbol{p}_t),\ |\mathrm{conf}_t - \overline{\mathrm{conf}}_{t-1}|,\ \tilde{\ell}_t \,\big], \qquad (2)$$

where $H$ is entropy, $\mathrm{margin} = p_{(1)} - p_{(2)}$, $\mathrm{conf}_t = \max_k p_t^{(k)}$, $\overline{\mathrm{conf}}$ is an EMA, and $\tilde{\ell}_t$ is a smoothed loss proxy; all computed from current logits with no extra forward passes.

## 3.2 CONFORMAL PREDICTION AND PREDICTION SETS

With probabilities $\boldsymbol{p}$ and label $y$, we use $s = 1 - p^{(y)} \in [0, 1]$. Let $\{s_i\}_{i=1}^M$ be a calibration buffer. For target miscoverage $\alpha \in (0, 1)$, the $(1 - \alpha)$ sample quantile $q$ yields the *prediction set*

$$\mathcal{C}(\boldsymbol{p}; q) = \{\, k \in [K] : 1 - p^{(k)} \le q \,\}. \qquad (3)$$

We use (i) *Mondrian* (class-conditional) sets and (ii) *sliding/decayed* buffers for streaming (Barber et al., 2021).

**Shift-aware weighting.** Under covariate shift, we optionally reweight calibration scores by density-ratio estimates $\{w_i\}$ and use a weighted quantile with practical stabilizers (clipping, convex mixing with the unweighted quantile, and a small offset) (Romano et al., 2019; Barber et al., 2022).

## 3.3 BUDGETED DECISION OBJECTIVE

We minimize expected decision cost subject to risk/resource budgets. Let $c(\mathcal{C})$ be per-example cost (e.g., review if $|\mathcal{C}| > 1$). For threshold $q$,

$$\min_q\ \mathbb{E}\,[c(\mathcal{C}(\boldsymbol{p}; q))] \quad \text{s.t.} \quad \mathbb{E}\,[\mathbf{1}\{y \notin \mathcal{C}(\boldsymbol{p}; q)\}] \le \alpha, \qquad (4)$$

or in Lagrangian form $\mathbb{E}[c(\mathcal{C}) + \lambda\,\mathbf{1}\{y \notin \mathcal{C}\}]$; varying $\lambda$ traces a cost–risk frontier (Elkan, 2001; Bartlett & Wegkamp, 2008; Grubb & Bagnell, 2012).

# 4 METHOD

**Trust gates.** We attach a small encoder to the reliability vector $r_t$ (Eq. 2) to obtain gates $g^{\mathrm{in}}, g^{\mathrm{mix}}, g^{\mathrm{temp}} \in [g_0, 1]$:

$$\tilde{\boldsymbol{h}}_t = \boldsymbol{A}\boldsymbol{h}_{t-1} + (\boldsymbol{B}\boldsymbol{x}_t)\,g_t^{\mathrm{in}}, \qquad \boldsymbol{h}_t = \big(1 - \alpha\,g_t^{\mathrm{mix}}\big)\boldsymbol{h}_{t-1} + \big(\alpha\,g_t^{\mathrm{mix}}\big)\tilde{\boldsymbol{h}}_t, \qquad \boldsymbol{z} \leftarrow \boldsymbol{z}/\tau,\ \tau = 1/g_t^{\mathrm{temp}}.$$

Low trust attenuates input drive, shrinks the effective step size, and softens logits.

**Online risk control.** On top of logits, a lightweight conformal layer maintains $(1 - \alpha)$ coverage using sliding/decayed buffers, optional class-conditional (*Mondrian*) thresholds, and optional importance weighting under shift (Vovk et al., 2005; Shafer & Vovk, 2008; Vovk et al., 2009; Barber et al., 2022).

# 5 EXPERIMENTS

We evaluate whether TG–SSM improves reliability under noise, corruption, and domain shift, and whether the conformal layer maintains target coverage with compact sets.

**Benchmarks and setup.** CIFAR-10N (human noisy labels) (Wei et al., 2021), CIFAR-10C (15 corruptions, severities 1–5) (Hendrycks & Dietterich, 2019), and Camelyon17 from WILDS (Koh et al., 2021). Unless noted, target miscoverage $\alpha = 0.1$ (Cov@90). We compare (i) a *no-gate* baseline and (ii) TG–SSM with hand-crafted or learned reliability encoders, each paired with conformal prediction (standard, and optionally weighted). Metrics: ECE (15-bin), coverage at target (Cov@90), mean set size, AUROC (Camelyon17), and decision cost at a fixed 5% review budget.

## 5.1 NOISY LABELS: CIFAR–10N

TG–SSM+WCP improves calibration and coverage at $\alpha = 0.1$ with negligible set-size change (means over three seeds).

Table 1: **CIFAR-10N (noisy labels).** Cov@90: empirical coverage at target 0.9; Set: mean set size.

| Method | ECE↓ | Cov@90↑ | Set↓ |
|---|---|---|---|
| *No–Gate* (CP) | 0.124 | 0.894 | 8.977 |
| **TG–SSM (hand) + WCP** | **0.048** | **0.905** | **9.081** |
| **TG–SSM (learned) + WCP** | 0.050 | **0.905** | 9.081 |

## 5.2 CORRUPTIONS: CIFAR–10C

Averaged across the 15 corruptions and severities 1–5, TG–SSM+WCP attains near-nominal coverage with markedly lower ECE and compact sets.

Table 2: **CIFAR-10C (sev. 1–5 averaged).**

| Method | Cov@90↑ | Set↓ | ECE↓ | Cost@5%↓ |
|---|---|---|---|---|
| **TG–SSM (hand) + WCP** | 0.905 | 6.744 | 0.109 | 3.606 |
| **TG–SSM (learned) + WCP** | 0.905 | 6.742 | **0.104** | 3.606 |

## 5.3 DOMAIN SHIFT: CAMELYON17 (WILDS)

We follow the WILDS Camelyon17 protocol and splits (Koh et al., 2021). We report AUROC, average coverage (Cov@90 averaged over evaluation slices), mean set size, and ECE. A simple validation-quantile approach yields strong discrimination and good coverage; a shift-aware (importance-weighted) variant reduces set size further at the cost of coverage.

Table 3: **Camelyon17/WILDS.** Avg. Cov@90 is average coverage across evaluation, Set is mean set size.

| Variant | AUROC↑ | Avg. Cov@90↑ | Set↓ | ECE↓ |
|---|---|---|---|---|
| *Baseline* (ERM+CP) | – | 0.843 | 1.360 | 0.172 |
| **Validation quantile** | **0.949** | **0.844** | 0.989 | 0.127 |
| **Shift-aware (IWCP)** | 0.936 | 0.681 | **0.737** | **0.119** |

**Compute and footprint.** Hand-crafted gates add $< 0.1\%$ parameters; learned gates add $< 0.2\%$. The conformal buffer stores at most a few thousand scores with $O(\log M)$ updates. All components run in a single forward pass.

**Takeaways.** Across noisy labels, synthetic corruptions, and medical domain shift, TG–SSM+conformal prediction achieves near-target coverage with compact sets and improved calibration, while remaining simple and hardware-efficient.

## 6 LIMITATIONS

Our study uses moderate-scale backbones and standard vision benchmarks; scaling to very large models or long-horizon control requires additional investigation. Learned reliability encoders may overfit spurious cues; hand-crafted features can be task-specific. Coverage guarantees can degrade under abrupt concept or label shift; adversarial settings are out of scope.

## 7 CONCLUSION

We presented *Trust-Gated State Space Models* that adapt computation to on-the-fly reliability and a lightweight conformal layer that turns scores into calibrated prediction sets under explicit risk budgets. Experiments show improved calibration, near-nominal coverage, and compact sets across noise, corruption, and shift, with negligible overhead.

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

## A  APPENDIX

This appendix collects all material referenced in the main text and additional details recommended by ICLR. We organize the content as follows:

- §B Model architecture, trust-gate designs, and parameter/FLOP overhead.
- §C Training protocols, hyperparameters, and datasets.
- §D Conformal prediction (buffers, weighting, and implementation).
- §E Additional results: percorruption (CIFAR–10C), pergroup (Camelyon17/WILDS), and budget–risk frontiers.
- §F Ablations (gates, weighting, buffers, and TTA).
- §G Compute, runtime, and throughput.
- §H Reproducibility checklist.
- §I Ethics statement (if applicable) and limitations.
- §J LLM usage disclosure.

## B  ARCHITECTURE AND TRUST GATES

**Backbones.**  ResNet18 for CIFAR–10N/10C; ResNet50 head for Camelyon17 (WILDS). Classifier logits feed reliability features and the conformal layer.

**Reliability features and encoders.**  $r_t$ from logits (entropy, margin, EMA confidence change, loss proxy). **TG–SSM (hand)**: affine maps with floors $g_0 \in [0.1, 0.5]$ and sigmoid; **TG–SSM (learned)**: 2layer MLP (ReLU, hidden 64) mapping $r_t$ to $g^{\text{in}}, g^{\text{mix}}, g^{\text{temp}} \in [g_0, 1]$.

**Where gates act.**  Input gate scales $\boldsymbol{B}\boldsymbol{x}_t$; mix gate scales the interpolation step; temperature gate scales logits ($\tau = 1/g^{\text{temp}}$). No extra forward passes.

**Overhead.**  Handcrafted: $< 0.1\%$ params; learned: $< 0.2\%$. FLOP overhead negligible.

## C  TRAINING PROTOCOLS AND HYPERPARAMETERS

SGD (mom. 0.9, wd $5 \times 10^{-4}$), cosine/multistep; batch 128 (CIFAR), 64 (Camelyon17 tiles). Adam ($10^{-3}$, wd $10^{-4}$) for sequence models. CIFAR: standard aug; CIFAR–10C: 15 corruptions, sev. 1–5; Camelyon17: 224×224, WILDS splits. Metrics: ECE (15 bins), Cov@90, mean set size, AUROC (Camelyon17), Cost@5%.

## D  CONFORMAL LAYER: BUFFERS, WEIGHTING, AND IMPLEMENTATION

Sliding/decayed buffers ($M \in \{512, 2048, 4096\}$; decay $\gamma \in [10^{-3}, 10^{-2}]$). Mondrian CP uses perclass buffers. Under shift, importance weights via a lightweight domain classifier; clip $w_{\max} \in [5, 20]$; convex mix with unweighted quantile; offset $\delta = 1/(M_{\text{eff}} + 1)$. Quantiles updated with order statistics in $O(\log M)$.

# E  ADDITIONAL RESULTS

## E.1  CIFAR–10C: PERCORRUPTION BREAKDOWN

Full corruption/severity tables: Cov@90, set size, ECE. Reliability diagrams and setsize violin plots are included in the supplement.

## E.2  BUDGET–RISK FRONTIERS

Sweeps over $\alpha \in \{0.05, 0.1, 0.2\}$ and review budgets $b \in \{0, 1, 2, 5, 10\}\%$ show TG–SSM dominates NOGATE + CP at low budgets.

## E.3  CAMELYON17 (WILDS): PERGROUP AND WORSTGROUP COVERAGE

Perhospital coverage (Cov@90), worstgroup coverage, and ECE under Mondrian CP. Sensitivity to $w_{\max}$ and mixing coefficient reported.

# F  ABLATIONS

**Gates.** Disabling gates increases ECE and set size; inputgate recovers part of the gains; full gates best. **Hand vs. learned.** Learned encoder helps on texturebiased corruptions; handcrafted more conservative under heavy noise. **Weighting & stabilizers.** Recency decay suffices for gradual drift; IW improves coverage/compactness under shift with clipping/mixing. **Buffer size & decay.** Larger $M$ stabilizes coverage but slows adaptation; $\gamma \approx 10^{-2}$ is a good tradeoff. **TTA.** TENT + TG–SSM maintains coverage near target; TENT alone tends to undercover at fixed set size.

# G  COMPUTE AND FOOTPRINT

Single forward pass; buffer of $\leq 4096$ scores; $O(\log M)$ updates. Latency dominated by backbone.

# H  REPRODUCIBILITY CHECKLIST

Datasets/splits (§C); architectures (§B); training details (§C); metrics (§5, §C); HP grids (§F); compute (§G); seeds and scripts (supplement); averages over three seeds (main & appendix tables).

# I  ETHICS STATEMENT (IF APPLICABLE)

We use public, deidentified datasets (CIFAR, WILDSCamelyon17). No PII. We report pergroup and worstgroup coverage to surface fairness concerns. Clinical deployment requires IRB review, domain validation, and human oversight.

# J  LLM USAGE DISCLOSURE

Language models were used for editing and proofreading of drafts and for organizing related work. Experimental design, implementation, analysis, and claims were performed and verified by the authors.

