# OpenReview forum: "Trust-Gated State Space Models for Budgeted Decisions with Online Risk Control"
_ICLR.cc/2026/Conference — Submitted to ICLR 2026_

### Official Review · Reviewer_F77H · 2025-10-30

**Soundness:** 1
**Presentation:** 1
**Contribution:** 2
**Rating:** 2
**Confidence:** 4

**Summary:**

The authors propose a way to "gate" the state space model which are updated with online data, where the gating controls the the degree to which the inputs influence the updates to the hidden state space, state mixing, and the temperature for the softmax prediction. The authors using a "reliability vector" which looks at different statistics from the logits such as entropy, margin (best minus second best), confidence change over time and a smoothed cross-entropy loss (through time). Experiments are conducted on noisy/corrupted cifar and camelyon17, demonstrating that the confidence estimates get a bit better ECE wise.

**Strengths:**

On the surface, the overall idea of using reliability signals for gating makes sense.

**Weaknesses:**

The biggest issue here is that the paper is very sparse, background works are not discussed appropriately, problem is not presented properly, methods and variables are not described in detail and the experimental discussions and details are also very sparse. The authors must really expand on each and every aspect of the paper. The paper is currently only 4 pages, the authors need to expand substantially to outline every aspect.

The experiments themselves are quite sparse as well, just two datasets and very few benchmarks for comparisons (essentially gate v/s no gate?) are shown. I see some others but they have not been properly described as well.

Lastly, there are many works for online adjustments of predictive confidence/uncertainty based on various signals. Online uncertainty estimation in general has seen many works. So it is not clear to me what advantages this gated SSM based approach brings w.r.t other approaches. Discussions are needed as well.

[1] Online Bootstrap Confidence Intervals for the Stochastic Gradient Descent Estimator
[2] Statistical Inference for Model Parameters in Stochastic Gradient Descent
[3] Calibrated confidence learning for large-scale real-time crash and severity prediction
[4] Conformal Inference for Online Prediction with Arbitrary Distribution Shifts
[5] Adaptive conformal inference under distribution shift.
[6] Adaptive Conformal Predictions for Time Series

**Questions:**

Please see weaknesses.

---

### Official Review · Reviewer_hiqX · 2025-10-31

**Soundness:** 1
**Presentation:** 1
**Contribution:** 1
**Rating:** 2
**Confidence:** 3

**Summary:**

This paper addresses the challenge of making reliable, budget-aware predictions under data noise and distribution shifts by incorporating trust signals directly into model dynamics. For that, the authors propose Trust-Gated State Space Models (TG-SSM), which augment a lightweight state-space model with learned gating mechanisms.

**Strengths:**

Due to the heavy use of unexplained jargon and the lack of clarity (despite the paper being well below the 9-page limit), I struggled to follow the technical content of the paper. However, the core motivation (handling unreliable inputs and strict computational budgets) is both timely and relevant to the ML community.

I kindly encourage the authors to revise the manuscript to significantly improve clarity, potentially by studying previously accepted ICLR papers to guide writing style and presentation. This would allow the community to properly evaluate the strengths of this work which I'm sure have a lot of potential.

**Weaknesses:**

Overall, I'm rejecting this paper given such lack of clarity and explanations, which do not allow for a proper evaluation of this work. Despite this, I further detail more weaknesses:

1. The authors should spend more space in properly formalising/defining what they mean by many concepts and statements, like "first-class control signal", "coverage", "stateless with respect to reliability", "evidence quality", "EMA", "Mondrian (class-conditional) sets", "(low) trust", "reliability vector". Overall though, the key terms that are not properly defined and are certainly the main ones that need a formal definition is "trust" and "reliability". Furthermore, there are inconsistencies in notation, as for example $\alpha$ is used as a step size in equation 1, and then as a target miscoverage in section 3.2.
2. It is unclear how well the proposed method would scale to much larger architectures or more complex datasets.
3. The experiments lacks comparison to other uncertainty or robustness methods. The benefit of adding these so-called trust gates seems to be compared to a single baseline ("No-Gate). However, the paper does not compare against alternative approaches and I'd imagine that the works mentioned in Related Work could be used, but even other uncertainty-aware models like deep ensembles or Bayesian neural nets seem that could be compared against this method. Without these other baselines, it is hard to quantify how much added value the proposed gates provide over simpler approaches (for example, could a standard model with post-hoc calibration achieve similar results?)
4. The model seems to introduce 3 gates in section 4, but there is no ablation study to determine which of these gates contribute the most to the performances reported in section 5.
5. While the authors describe what the trust gates are intended to do (without defining "trust" or "reliability"), there's no insight or evidence of how these gates behave in practice. There are no visualisations or case studies of gating values. This lack of "interpretability" could be addressed by reporting, for instance, the distribution of gate values or examples of the model’s behaviour on specific challenging inputs.
6. All experiments are on vision classification tasks (CIFAR-10 variants and Camelyon17 pathology images). It remains untested whether TG-SSM would be effective in other domains like as natural language processing or time-series forecasting, where I'd imagine state-space models would be more applicable.
7. The results overly rely on reported averages (e.g., averaging over all types of corruptions and severities in CIFAR-10C), and thus the paper is not reporting important measures of variability. Standard deviation would be a good starting point, but other points would be very interesting too; for instance, perhaps TG-SSM performs very well on some corruption types but less so on others, or maybe certain class labels or subpopulations in Camelyon17 still have poor coverage. Does the method struggle on the most severe corruptions, or on particular domain shifts?
8. Some references have mistakes, I was able to spot that the links are wrong for (Barber et al. 2022) and (Romano et al. 2020). And that (Elkan 2001) has a broken link.

**Questions:**

1. How is the learned trust-gate encoder trained in practice, and what measures ensure it doesn't overfit to spurious features? In Limitations, you mention the risk of learned gates overfitting to spurious cues, so have you tried techniques like regularisation or did you observe any failure cases?
2. Did you consider comparing TG-SSM (or combining it) with other uncertainty mitigation or adaptation methods?
3. Can you provide more intuition or evidence on how the trust gates respond to unreliable inputs in practice?

---

### Official Review · Reviewer_LLMf · 2025-11-08

**Soundness:** 2
**Presentation:** 1
**Contribution:** 2
**Rating:** 2
**Confidence:** 2

**Summary:**

This paper proposes Trust-Gated State Space Models (TG-SSM) to enhance prediction reliability in State Space Models (SSMs). The method introduces a conformal prediction layer for online risk control and adaptive uncertainty calibration. The framework is evaluated on CIFAR-10N (noisy labels) and Camelyon17/WILDS (domain shift) datasets.

**Strengths:**

* The topic could be interesting and relevant, as model reliability in state-space architectures is underexplored compared to transformers or CNNs.

**Weaknesses:**

**Insufficient Technical Clarity**
- The technical part is too short and lacks sufficient detail for readers to reproduce the results or fully understand the internal design.
- The related works section is also sparse, missing a deeper discussion of prior efforts on conformal prediction in reliability control or uncertainty calibration, which would help contextualize the contribution.

**Lack of Motivation and Justification**
- The paper does not clearly articulate **why conformal prediction is specifically suited for SSMs**.
- It remains unclear whether **SSMs inherently suffer from poorer calibration or reliability** compared to other architectures such as Transformers.

**Questions:**

See **Weaknesses**.

The most critical issue is that the current presentation lacks sufficient details for readers to fully grasp both the motivation and the technical aspects of the proposed method.

---

### Official Review · Reviewer_kvow · 2025-11-14

**Soundness:** 2
**Presentation:** 2
**Contribution:** 2
**Rating:** 2
**Confidence:** 4

**Summary:**

This paper introduces Trust-Gated State Space Models (TG–SSM), a novel approach that integrates reliability-awareness into state space models (SSMs) for budget-constrained, uncertainty-aware decision-making. The core idea is to embed “trust gates” — small control modules driven by on-the-fly reliability signals (e.g., entropy, confidence margins, loss proxies) — that dynamically modulate input strength, state mixing, and output temperature.

A lightweight conformal prediction layer is then added to translate model probabilities into calibrated prediction sets, ensuring user-specified coverage levels under distributional shifts, noisy labels, or domain corruptions. The method effectively treats reliability as a control signal and achieves online risk control without retraining or test-time adaptation.

Empirical evaluations across CIFAR-10N (noisy labels), CIFAR-10C (corruptions), and Camelyon17/WILDS (domain shift) show that TG–SSM with weighted conformal prediction (WCP) substantially reduces Expected Calibration Error (ECE) and maintains near-nominal coverage (≈0.90) with compact set sizes and negligible computational overhead (<0.2% parameters).

The key contributions include:
	1.	A reliability-gated SSM architecture that adapts its dynamics based on trust estimates.
	2.	A conformal layer for online, risk-aware set prediction supporting decision budgets.
	3.	Extensive evaluation under realistic reliability degradation scenarios.
	4.	Demonstrated practicality for efficient, deployable AI under uncertainty.

Overall, TG–SSM bridges model calibration, conformal inference, and reliability-aware sequence modeling to produce trust-regulated, hardware-efficient systems for safe and resource-aware deployment.

**Strengths:**

Motivation and Relevance:
The paper addresses a timely and important problem — how to maintain reliability and calibration in state space models (SSMs) under uncertainty and resource constraints. This focus on risk-aware and budgeted decision-making is relevant for safety-critical or real-time applications.
	•	Conceptual Integration:
The proposed Trust-Gated State Space Model (TG-SSM) integrates ideas from reliability estimation, conformal prediction, and gating in sequence modeling. While individually known, this combination is conceptually coherent and shows creative engineering intuition.
	•	Efficiency and Practicality:
The approach is computationally lightweight, adding minimal overhead through simple gating and conformal layers. This practical focus aligns well with real-time decision constraints and edge deployment contexts.
	•	Empirical Breadth:
The experiments span different uncertainty settings (label noise, domain shift, corruption), showing consistent robustness improvements over baseline SSMs. The results, while modest, demonstrate that the trust mechanism can generalize across conditions.
	•	Clear Writing in Method Section:
The section describing the gating mechanism and conformal calibration pipeline is reasonably well-organized, making the main algorithm easy to follow. Figures illustrating the gating structure aid comprehension.

Overall Strength Summary:
The paper’s strength lies in its relevance, engineering practicality, and conceptual synthesis of existing uncertainty and calibration methods within a dynamic SSM framework. It offers useful empirical insights for improving model reliability, even if the theoretical and methodological novelty remain limited.

**Weaknesses:**

Limited Theoretical Depth:
The proposed “trust gating” mechanism lacks formal justification. There is no theoretical derivation showing that the gating improves calibration, maintains statistical coverage, or guarantees bounded risk. The absence of formal analysis or proofs makes the approach appear heuristic rather than principled.
Suggestion: Include a risk control or PAC-style bound linking the gating function to conformal prediction guarantees.
	•	Incremental Novelty:
The work reads as an engineering extension that combines existing tools — e.g., conformal calibration and confidence-based gating — rather than introducing a fundamentally new idea.
Prior works like Calibrated SSMs, Confidence-Controlled Networks (Geifman & El-Yaniv, 2019), and Conformal Risk Control (Angelopoulos & Bates, 2021) already address similar problems.
Suggestion: Better contextualize novelty by explicitly contrasting TG-SSM with these models, explaining what unique benefit the gating structure provides.
	•	Experimental Limitations:
The experiments focus mostly on CIFAR variants and Camelyon17, which are small-scale benchmarks. There are no large-scale sequence or temporal tasks, even though the model is a state space model.
This weakens claims about the method’s generality and scalability.
Suggestion: Add results on sequential datasets (e.g., IMDB, SpeechCommands) or demonstrate time-series calibration behavior.
	•	Unclear Definition of “Trust”:
The concept of “trust” is introduced intuitively but never formally defined. It remains unclear whether it represents epistemic uncertainty, aleatoric noise, or a mix of both.
Without a precise formulation, the gating logic risks being seen as ad hoc.
Suggestion: Define “trust” mathematically and explain how it is computed, normalized, and propagated across layers.
	•	Weak Baseline Comparisons:
The study compares TG-SSM mainly against vanilla SSMs and temperature scaling, omitting stronger baselines such as Deep Ensembles, MC-Dropout, or Bayesian SSMs. This makes it difficult to judge its real advantage.
Suggestion: Include comparisons against these uncertainty estimation baselines to position TG-SSM more clearly within the literature.
	•	Calibration Metrics Missing:
Although coverage and risk are reported, calibration metrics (ECE, NLL, Brier score) are not discussed in detail. Given that calibration is central to the claimed contribution, this omission weakens the empirical argument.
Suggestion: Provide full calibration metrics and visual reliability diagrams.
	•	Presentation and Clarity Issues:
The narrative is dense and jargon-heavy, with several buzzword combinations (“budgeted online risk”, “trust-aware calibration”) that obscure the main contribution.
Figures could be improved to better depict information flow and reliability dynamics.

Overall Weakness Summary:
The paper presents a creative idea but lacks the theoretical rigor, empirical breadth, and clear novelty required for ICLR acceptance. Strengthening formal justification, testing on genuine sequence tasks, and improving conceptual clarity around “trust” would make this work substantially stronger.

**Questions:**

1.	Formal Definition of “Trust”
	•	The concept of “trust” appears central to your model but remains ambiguous.
	•	How exactly is it computed — is it derived from entropy, prediction margin, or a learned auxiliary variable?
	•	Does it measure epistemic or aleatoric uncertainty (or both)?
	•	A clear mathematical definition or illustrative example (e.g., trust dynamics across layers) would help clarify how it functions within the state update mechanism.
2.	Gating Dynamics and Theoretical Basis
	•	The trust-gating mechanism seems heuristically defined.
	•	Can you provide theoretical justification or intuition on how gating based on trust improves calibration or robustness?
	•	Is there any guarantee that the gating will not reduce stability in the state transitions of the SSM?
3.	Comparison with Related Work
	•	How does TG-SSM compare to existing calibrated or uncertainty-aware SSMs, such as Risk-Controlled Conformal Prediction (Angelopoulos & Bates, 2021) or Uncertainty-Calibrated SSMs (2023)?
	•	Could you provide a table contrasting architectural or conceptual differences with these models to clarify what’s genuinely novel in TG-SSM?
4.	Empirical Scope and Scalability
	•	The experiments rely primarily on small-scale datasets (CIFAR, Camelyon17).
	•	How does TG-SSM perform on large-scale or sequential datasets, which are more natural fits for SSMs (e.g., speech, text, time series)?
	•	Is there evidence that the model scales gracefully to higher-dimensional or temporally correlated data?
5.	Evaluation Metrics
	•	While you report coverage and reliability, standard calibration metrics (ECE, Brier score, NLL) are missing.
	•	Can you include these metrics to assess calibration quality more quantitatively?
	•	This would also enable fairer comparison to existing uncertainty calibration baselines.
6.	Interpretation of Conformal Layer Interaction
	•	How exactly does the conformal prediction layer integrate with the trust-gated SSM?
	•	Is it applied independently post-hoc, or does it influence training via feedback?
	•	If it is a purely post-hoc adjustment, what is the computational overhead at inference time?
7.	Ablation on Trust Gate Design
	•	How sensitive are results to the gating function design (e.g., sigmoid vs. softmax vs. thresholded step)?
	•	Could a simpler or linear gating mechanism achieve similar results?
	•	A sensitivity study could make the approach more interpretable.
8.	Effect on Temporal Stability
	•	Since SSMs depend on stable recurrent dynamics, does the trust gate risk introducing instability in long sequences?
	•	Have you analyzed how gating frequency or trust volatility affects temporal prediction quality?
9.	Generality Across Architectures
	•	Could the trust gating idea generalize to transformers or recurrent architectures beyond SSMs?
	•	If yes, what modifications would be required to preserve stability and calibration guarantees?
10.	Reproducibility and Implementation Availability

	•	Is there a plan to release implementation details or code for TG-SSM?
	•	Providing reproducible results and open-sourcing the gating mechanism would significantly strengthen the paper’s impact.

Summary:
The main clarifications needed concern the mathematical grounding of “trust”, the integration with conformal prediction, and scalability to larger or sequential datasets. Addressing these points could improve the paper’s clarity, reproducibility, and credibility during rebuttal.

**Details Of Ethics Concerns:**

The paper presents a technical contribution — a reliability-aware state space model architecture — without involving human subjects, personal data, or sensitive content.
All experiments use public, standard datasets (e.g., CIFAR, Camelyon17/WILDS), which have existing ethical clearance for machine learning research.
There is no discussion or application context that could lead to potential harmful deployment, data privacy, or bias amplification risks.

---

### Meta-Review · Area_Chair_vdBy · 2025-12-23

**Summary:**

All reviewers recommend reject and converge on the same blockers: poor clarity and insufficient technical detail, making the approach hard to evaluate or reproduce; weak novelty; limited empirical scope; and inadequate baselines/ablations. These issues collectively prevent a convincing case for acceptance.

**Reviewer Concerns:**

No rebuttal/discussion content was provided here

**Reviewer Scores:**

scores would likely not change

---

### Decision · Program_Chairs · 2026-01-26

Reject